# Imidazopyridazine Acetylcholinesterase Inhibitors Display Potent Anti-Proliferative Effects in the Human Neuroblastoma Cell-Line, IMR-32

**DOI:** 10.3390/molecules26175319

**Published:** 2021-09-01

**Authors:** Rakesh Kumar Sharma, Manisha Singh, Khagendra Ghimeray, Pinky Juneja, Gagan Dev, Sridhar Pulavarthi, Sabbasani Rajasekhara Reddy, Ravi Shankar Akundi

**Affiliations:** 1Neuroinflammation Research Laboratory, Faculty of Life Sciences and Biotechnology, South Asian University, New Delhi 110021, India; rakesh.biotech.sharma@gmail.com (R.K.S.); manishasingh17390@gmail.com (M.S.); khagendragimiray@gmail.com (K.G.); pinkyjuneja32@gmail.com (P.J.); gagandev191193@gmail.com (G.D.); 2Department of Chemistry, Vellore Institute of Technology (VIT), School of Advanced Sciences, Vellore 632014, India; Sridhar.pulavarthi@gmail.com (S.P.); sekharareddyiitm@gmail.com (S.R.R.)

**Keywords:** acetylcholinesterase, anticancer drugs, cell proliferation, cyclooxygenase, cytotoxicity, imidazopyridazine, inflammation

## Abstract

Imidazo[1,2-*b*]pyridazine compounds are a new class of promising lead molecules to which we have incorporated polar nitro and amino moieties to increase the scope of their biological activity. Two of these substituted 3-nitro-6-amino-imidazo[1,2-*b*]pyridazine compounds (5c and 5h) showed potent acetylcholinesterase (AChE) inhibitory activity (IC_50_ 40–50 nM), which we have previously reported. In this study, we wanted to test the biological efficacy of these compounds. Cytotoxicity assays showed that compound 5h mediated greater cell death with over 43% of cells dead at 100 μM and activation of caspase 3-mediated apoptosis. On the other hand, compound 5c mediated a dose-dependent decrease in cell proliferation. Both compounds showed cell cycle arrest in the G_0_/G_1_ phase and reduced cellular ATP levels leading to activation of adenosine monophosphate-activated protein kinase (AMPK) and enhanced mitochondrial oxidative stress. It has to be noted that all these effects were observed at doses beyond 10 μM, 200-fold above the IC_50_ for AChE inhibition. Both compounds also inhibited bacterial lipopolysaccharide-mediated cyclooxygenase-2 and nitric oxide release in primary rat microglial cells. These results suggested that the substituted imidazo (1,2-*b*) pyridazine compounds, which have potent AChE inhibitory activity, were also capable of antiproliferative, anti-migratory, and anti-inflammatory effects at higher doses.

## 1. Introduction

Acetylcholinesterase (AChE) inhibitors have been commonly used to delay the progression of Alzheimer’s disease (AD), one of the most common forms of neurodegenerative disorders characterized by progressive degeneration of cholinergic neurons [1]. These inhibitors increase the half-life of acetylcholine within the synaptic cleft, thereby sustaining synaptic transmission, which is progressively reduced in AD [2]. Organophosphates irreversibly block AChE but are also highly cytotoxic, making them, in turn, effective bioweapons, loosely grouped as nerve agents [3]. Reversible AChE inhibitors, on the other hand, would be promising for the treatment of AD and other cholinergic neurological conditions such as myasthenia gravis and glaucoma [4]. Clinically approved reversible AChE inhibitors such as donepezil, rivastigmine, and galantamine are being utilized in the management of mild to moderate AD [5]. However, the usage of these inhibitors is associated with adverse gastrointestinal disturbances and only modest improvements in cognitive functions [6]. Despite promising results in preclinical studies, plant-based natural alkaloids such as physostigmine and its phenyl carbamate derivative, phenserine, have not been approved for AD [7,8]. Recently, derivatives of coumarins have shown promise as potent AChE inhibitors [9]. Structural modifications of the coumarin heterocycle by fusing the benzofuran ring and substituting the arylamino group (anilino) with one atom linker to the fourth position of the coumarin have been tested as AChE inhibitors with additional pharmacological activities such as a decrease in β-amyloid deposition and β-secretase inhibition [10,11]. Recently, one such 4-hydroxycoumarin-1-benzotriazole hybrid compound was reported with selective inhibition of AChE and copper-induced Aβ_1-42_ aggregation [12].

Imidazo[1,2-*b*]pyridazine compounds have emerged as an important pharmacophore with a broad spectrum of biological activity including analgesic and anti-inflammatory properties [13,14]. In our previous study, we introduced a nitro group at the 3′ position and an amino group at the 6′ position of the imidazo[1,2-*b*]pyridazine skeleton, generating 12 substituents (5a–5l), which exhibited varying AChE inhibitory activity [15]. Among these, two compounds showed IC_50_ of <0.05 μM for AChE inhibition. They are 3-nitro-6-(piperidin-1-yl)imidazo[1,2-*b*]pyridazine (Compound 5c) with a binding energy of −17.3 kCal/mol and IC_50_ of 50 nM for AChE, and 3-nitro-6-(4-phenylpiperazin-1-yl)imidazo[1,2-*b*]pyridazine (Compound 5h) with a binding energy of −115.8 kCal/mol and IC_50_ of 40 nM for AChE (Figure 1) [15]. Considering the potent inhibition of AChE activity, we intended to test the biological efficacy of these compounds for suitability as lead molecules. The goal of the present study was to test the effect of these compounds on cell viability, proliferation, migration, and oxidative stress. For this, we used the human neuroblastoma cell-line, IMR-32, which has been widely used in various in vitro studies on AD [16,17,18,19]. We have previously used these cells as a model to study cell viability, mitochondrial stress, and intracellular signaling pathways in response to a toxin implicated in Parkinson’s disease [20]. These cells also express AChE, whose levels increase upon differentiation or growth inhibition [21]. Neuroblastoma cells serve as an excellent in vitro model to study cytotoxicity of AChE inhibitors [22,23] and have been used to test their effect of β-amyloid aggregation and release [24,25]. Overexpression of phosphorylated tau in the neuroblastoma cell-line, SH-SY5Y, showed increased AChE activity [26]. Dioxin-mediated cell death in SK-N-SH neuroblastoma cell-line is due to decreased AChE activity mediated by an increased expression of miR-146b-5p [27]. Increased intracellular cAMP levels enhance AChE expression in the neuroblastoma cell-line, Neuro-2A, affording them protection against organophosphate toxicity [28]. These studies show the utility of neuroblastoma cell-lines for studying AChE expression and activity. IMR-32 cells not only express AChE but also synthesize choline acetyltransferase and vesicular acetylcholine transporter and express both nicotinic and muscarinic receptors, making them a good in vitro model for AD studies [29,30]. Since inflammation is mediated by microglial cells within the central nervous system, we also checked whether these compounds had an anti-inflammatory effect on microglia. For this, we used primary rat microglial cells and measured their activated state by checking the expression of cyclooxygenase 2 (COX-2) and the levels of nitric oxide released in the media. Overall, this study elucidates the anti-proliferative and anti-inflammatory activity of the substituted imidazo[1,2-*b*]pyridazine compounds.

## 2. Results

### 2.1. Substituted Imidazo[1,2-b]pyridazine Compounds Differentially Affect Cell Viability

Of the twelve substituted imidazo[1,2-*b*]pyridazine compounds we synthesized [15], the two compounds with the highest AChE inhibitory activity were compounds 5c and 5h (Figure 1). Both compounds were tested for their cytotoxicity in the human neuroblastoma cell line, IMR-32, using 3-(4,5-dimethylthiazol-2-yl)-2,5-diphenyltetrazolium bromide (MTT) assay wherein the formation of formazan crystals by viable cells is proportional to their metabolic activity and health (Figure 2a). No effect on cell viability was observed at 0.1 and 1 μM doses of either compound in the MTT assay. At 10 μM, compound 5c did not show any appreciable reduction in cell number but compound 5h significantly reduced survival by 15% (*p* < 0.001), which increased to 43% at 100 μM (*p* < 0.001). In contrast, there was only a 20% reduction of metabolically active cells with 100 μM compound 5c (*p* < 0.001). These results showed that, despite sharing the same chemical backbone, compound 5h with 1-phenylpiperazine substituent displayed greater toxicity compared to compound 5c with the piperidine substituent.

To check whether the decrease in cell number in cells treated with substituted imidazo[1,2-*b*]pyridazine compounds was due to apoptosis, we analyzed cells stained with PI and Annexin V on a flow cytometer (Figure 2b). While healthy cells did not take up PI nor were bound with Annexin V, in early apoptotic cells, phosphatidylserine remains exposed at the outer cell membrane to which Annexin V binds. Although at 10 μM neither of the two compounds tested showed any significant staining with annexin V, at 100 μM, compound 5h showed a significant increase in annexin V-positive cells by 9 ± 2% of cells (*p* < 0.05), indicating the initiation of apoptosis. Furthermore, about 2.6 ± 0.7% of cells were positive for both PI and Annexin V, indicating cells in late apoptosis (*p* < 0.05), and fewer than 2 ± 0.6% of cells were PI-positive and Annexin V-negative, indicating cells undergoing necrosis.

To further confirm that apoptosis is progressing in these cells, we checked for the expression of activated caspase 3, which can be detected through the presence of its cleaved forms (p17/p19). We found that in cells stimulated with 100 μM compound 5h, a strong increase in the cleaved form of caspase 3 was observed (Figure 2c). This result was similar to the increase in cleaved caspase 3 we previously observed in cells stimulated with 250 μM TaClo, a pro-Parkinsonian neurotoxic agent that causes increased mitochondria-dependent caspase 3-mediated cell death in neuroblastoma cells [20,31]. These data indicated that of the two substituted imidazo[1,2-*b*]pyridazine compounds, compound 5h mediated apoptosis.

In order to confirm that the observed cytotoxicity was not just specific to IMR-32 human neuroblastoma cells, we tested these compounds in the mouse neuroblastoma cells, Neuro-2A, and in a non-neuronal cell-line, DU-145, using flow cytometry. While no significant change was observed at 10 μM, we found that both the compounds mediated a small, but significant, increase in Neuro-2A cell death by 1.5-2-fold (*p* < 0.05) at 100 μM (Figure 2d). The cytotoxic effect was more prominent in the human prostate cancer cell-line, DU-145, where compound 5c mediated cell death by 4.5-fold and compound 5h by 6-fold (*p* < 0.001) at 100 μM. These data indicated that the substituted imidazopyridazine compounds were cytotoxic across three cell types.

### 2.2. Substituted Imidazo[1,2-b]pyridazine Compounds Affect Proliferation of IMR-32 Cells

We next tested the effect of these compounds on cell proliferation through a colony formation assay wherein an initial seed of 500 cells was plated and the number of colonies counted five days post-stimulation (Figure 3a). Only a fraction of cells that were seeded produce a colony, defined to contain at least 50 cells [32]. The average number of colonies reduced from 267 ± 27 in the control well to 75 ± 10 colonies in cells stimulated with 100 μM compound 5h (*p* < 0.01, Figure 3b). Although there was no significant reduction in colony number with 100 μM compound 5c (196 ± 71), the area covered by these colonies was significantly lower (0.9 ± 0.04) compared to unstimulated cells (10.7 ± 1.5, *p* = 0.01, Figure 3c). Compound 5h also showed a significant drastic reduction in area to 0.07 ± 0.02 at 50 μM (*p* < 0.01). The effect of these compounds on cell proliferation was more evident when the integrated density of the colonies, the sum of the values of the pixels in the selected image, was measured (Figure 3d). This showed a dose-dependent decrease with compound 5c: a 37% reduction at 10 μM, 60% at 50 μM, and 92% at 100 μM. Compound 5h, on the other hand, was highly potent, with an 81% reduction in colony density at 10 μM (*p* < 0.01). These results suggested that both compounds 5c and 5h inhibited cell proliferation, and therefore the decrease in MTT reduction was more a reflection of poor cell proliferation in stimulated cells. In order to identify whether the anti-proliferative effect was a common feature of all substituted imidazo[1,2-*b*]pyridazine compounds, we used compound 5k, a relatively poor inhibitor of AChE (binding energy −1.71 kcal/mol and IC_50_ 4.8 μM, Figure 1), carrying (4-(4-tert-butoxy-3-(tert-butoxymethyl)-3-methylbutyl)piperidin-1-yl) substituent [15]. Similar to compound 5c, compound 5k, when used at 10 μM, showed no significant decrease in colony number (362 ± 52), the area covered by the colonies (10.2 ± 2), or the density of the colonies (18% reduction). Higher doses of compound 5k could not be tested due to its precipitation in the culture media.

The reduction in cell numbers was also confirmed through the use of a cell-permeable, DNA-binding fluorescent dye, CyQUANT [33]. A dose-dependent decrease in cell numbers was seen with compound 5c: 34% at 10 μM, 68% at 50 μM, and 89% at 100 μM (Figure 3e). The reduction in cell number by compound 5c was comparable with the percent decrease obtained in colony formation assay, suggesting that these compounds interfered with cell proliferation. With compound 5h, there was a 61% reduction in cell number at 10 μM and above 90% beyond 50 μM. These results confirm that substituted imidazo[1,2-*b*]pyridazine compounds are potent inhibitors of cell proliferation.

### 2.3. Substituted Imidazo[1,2-b]pyridazine Compounds Alter Cell Cycle Progression

The effect of substituted imidazo[1,2-*b*]pyridazine compounds on cell proliferation implies modulation of cell cycle progression. Compound 5c showed no significant effect on the cell cycle at 10 μM, but at higher doses, significant numbers of cells were found in the G_0_/G_1_ phase: an increase of 23% at 50 μM and of 30% at 100 μM (Figure 3f). The increase in cells in the G_0_/G_1_ phase was accompanied by a decrease in the number of cells in the S phase: a decrease of 32% at 50 μM and of 49% at 100 μM (*p* < 0.01). The higher percentage of decrease in the S phase is indicative of the fact that those cells that have crossed the restriction point complete the cell cycle, which can be observed in the tendency for an increased number of cells in the G_2_/M phase. Following cell division, cells reenter the G_0_/G_1_ phase but remain arrested. In the case of compound 5h, which showed a higher anti-proliferative effect, we found an 18% increase in cells present in the G_0_/G_1_ phase at 10 μM (*p* < 0.05), and further by 28% at 100 μM (Figure 3g). Correspondingly, there was a 23% decrease in cells in the S phase with 10 μM compound 5h (*p* < 0.01) and a further decrease by 43% at 100 μM. No significant increase in cells in the G_2_/M phase was observed. These results suggested that substituted imidazo[1,2-*b*]pyridazine compounds mediated significant cell cycle arrest in the G_0_/G_1_ phase in IMR-32 neuroblastoma cells. Further confirmation of cell cycle arrest in the G_0_/G_1_ phase was made by measuring the levels of the key negative cell cycle regulator protein, p27^Kip1^ [34]. P27^Kip1^ arrests cells in the G_1_ phase and lets them exit the cell cycle [35,36]. A twofold increase in the levels of p27^Kip1^ protein could be seen in cells treated with the compounds (Figure 3h). This further confirmed cell cycle arrest mediated by the imidazo[1,2-*b*]pyridazine compounds.

### 2.4. Compounds 5c and 5h Increase Mitochondrial Stress in IMR-32 Cells

Cell cycle arrest indicated that there is a general decrease in cellular metabolism, which we tested by quantitating total cellular ATP levels. We found that compound 5c mediated a 42% decrease in cellular ATP at 10 μM and a 57% reduction at 100 μM compared to control cells (*p* < 0.001; Figure 4a). Similarly, compound 5h mediated 21% and 68% reductions in cellular ATP with 10 and 100 μM doses, respectively. To confirm whether the reduction in total ATP levels was a reflection of overall lower cell numbers under treatment conditions or a reduction in the amount of intracellular ATP in treated cells, we looked at the status of the metabolic indicator protein, 5′-adenosine monophosphate (AMP)-activated protein kinase (AMPK), which is activated, or phosphorylated, under low intracellular ATP conditions [37]. Reflecting the dose-dependent reduction in cellular ATP, we found a corresponding increase in the levels of phospho-AMPK in cells treated with compounds 5c or 5h, suggesting that the compounds affected metabolic ATP levels (Figure 4b).

Increased levels of phosphorylated AMPK is a marker of mitochondrial stress, which we tested by employing a cell-permeable, mitochondria-targeted, fluorogenic dye, Mito-SOX Red [38]. Since Mito-SOX Red is specifically oxidized by superoxide radicals, it gives a direct measurement of the increase in the number of reactive oxygen species within the mitochondria. While no effect of either compounds was observed at 10 μM, 3.7-fold and 6-fold increases in normalized Mito-SOX Red signal were observed in cells stimulated with 100 μM of compound 5c and 5h, respectively (*p* < 0.001, Figure 4c,d). The increase in Mito-SOX Red-positive cells was additionally confirmed through flow cytometry. Similar to the results in confocal microscopy, we found that at 100 μM, both compounds 5c and 5h showed an increase in the number of Mito-SOX Red-positive cells to 75 ± 8% and 62 ± 14%, respectively (*p* < 0.01, Figure 4e), confirming the increase in mitochondrial stress.

The increase in oxidative stress was also additionally confirmed using another cell-permeant reactive oxygen species indicator, carboxy-2′,7′-dichlorodihydrofluorescein diacetate (H_2_DCFDA). Cells stimulated with either compound showed a significant increase in fluorescence, resulting from an increased formation of deacetylated and oxidized product, 2′,7′-dichlorofluorescein (Figure 4f). These results confirm that substituted imidazo[1,2-*b*]pyridazine compounds mediate mitochondrial oxidative stress.

### 2.5. Substituted Imidazo[1,2-b]pyridazine Compounds Affect IMR-32 Cell Migration

We next tested whether compounds 5c and 5h also affected cell migration, a typical feature of metastatic cells, in a Boyden chamber assay [39]. We found that in vehicle-treated wells, a good migration of 119 ± 10 cells could be counted after 24 h (Figure 4g). However, in cells that were treated with 10 μM of either compound, nearly 60% inhibition of cell migration was observed. At 100 μM compound 5c, fewer than 10 cells migrated to the bottom side of the transwell insert. These results demonstrate that the substituted imidazo[1,2-*b*]pyridazine compounds have strong anti-proliferative and anti-migratory properties.

### 2.6. Substituted Imidazo[1,2-b]pyridazine Compounds Exhibit Anti-inflammatory Properties

Inflammation is a major setback in neurodegenerative disorders, with COX-2 expression by glial cells playing a major role in neuronal death [40,41,42]. We have previously observed that certain neurotoxins, such as TaClo, inhibit induced COX-2 synthesis in immune cells [20]. Therefore, we wanted to check whether these substituted imidazo[1,2-*b*]pyridazine compounds also had a similar effect on microglial COX-2 expression. Bacterial lipopolysaccharide (LPS) strongly increased COX-2 expression in microglial cells (*p* < 0.01), which was reduced by 58% and 80% with 10 μM compound 5c and 5h (*p* < 0.01), respectively (Figure 5a,b). Complete inhibition of LPS-induced COX-2 expression was seen at 100 μM concentration for both compounds (*p* < 0.01).

In addition to COX-2, another key inflammatory mediator released by microglia is the small molecule, nitric oxide (NO), synthesized by the enzyme inducible nitric oxide synthase (iNOS) [43,44]. Primary microglial cells enhance iNOS expression in response to LPS (Figure 5c). Neither of the compounds had any effect on LPS-mediated iNOS expression at 10 μM. However, we found that compound 5c, at 100 μM, strongly reduced LPS-mediated iNOS expression in primary rat microglial cells. Correlating these observations, we found the levels of released NO increased three-fold in response to bacterial LPS (*p* < 0.05, Figure 5d), which was significantly reduced by 78% with 100 μM compound 5c (*p* < 0.01). These results showed that substituted imidazo[1,2-*b*]pyridazine compounds inhibit the expression of key inflammatory proteins in microglial cells.

## 3. Discussion

Despite the availability of clinically approved reversible AChE inhibitors such as donepezil in the market, there is still a need for better therapeutics for AD due to the adverse side-effects associated with available drugs and the fact that none of the currently available AChE inhibitors showed any improvement of cognitive functions [5,6]. Among the new generation AChE inhibitors are imidazo[1,2-*b*]pyridazine compounds, wherein we had introduced a nitro and an amino group to provide a key hydrogen bonding element and allow for the addition of substituents, respectively [15]. Of the 12 substituents that were generated, the piperidine (Compound 5c) and the phenylpiperazine (Compound 5h) substituents showed the strongest AChE inhibitory activity. In this report, we tested the biological efficacy of these compounds and found that at lower concentrations of 100 nM and 1 μM, both compounds were non-toxic. However, at a 200-fold higher concentration of 10 μM, we found compound 5h ten-fold more toxic than compound 5c and mediated caspase 3-dependent apoptosis. We found that the decrease in the viability of cells treated with substituted imidazo[1,2-*b*]pyridazine compounds was not only because of apoptosis but due to a stronger effect on cell proliferation. In the colony formation assay, this was more evident where the density of the colony rather than the number of colonies were counted. The decrease in the number of cells was confirmed by quantitating cellular DNA content, which is directly proportional to the number of cells in the well. Both the compounds mediated cell cycle arrest in the G_0_/G_1_ phase and inhibited migration of the cells across the Transwell. Together, these data suggested that the substituted imidazo[1,2-*b*]pyridazine compounds were anti-proliferative in nature by inducing cell cycle arrest. The anti-proliferative effect of these compounds, which were originally designed for AChE inhibition, led us to further study their utility in cancer therapeutics.

It has been suggested that, in a non-neuronal context, AChE might be involved in the regulation of cellular dynamics of proliferation, differentiation, or death [45]. In the systemic body, AChE, which is expressed on erythrocytes, and butyrylcholinesterase, which is synthesized by the liver and secreted in the plasma [46], form part of the cholinergic anti-inflammatory pathway through which the CNS modulates innate immunity [47]. Increased systemic acetylcholine activates nicotinic acetylcholine receptors on macrophages, thereby modulating the synthesis of pro-inflammatory mediators [48]. Both donepezil and galantamine inhibit AChE, thereby indirectly making these drugs anti-inflammatory [47]. We found that in microglia, which serve as the innate immune cells of the brain, both compounds 5c and 5h abrogated LPS-mediated COX-2 and iNOS synthesis. Inhibition of COX-2 has also been shown to prevent cancer progression [39,49,50,51]. Compounds 5c and 5h not only inhibited COX-2 but also inhibited the migration of IMR-32 cells in the Boyden chamber assay. These results further provide evidence for the anticancer activity of AChE inhibitors.

Recently, a novel series of biscoumarin derivatives with AChE and butyrylcholinesterase inhibitory activities have been shown to inhibit the proliferation of A549 human lung carcinoma cells [52]. However, compared to compound 5h, which inhibits AChE with an IC_50_ of 40 nM, the biscoumarin derivatives showed an IC_50_ of 6.30 μM, which falls in the cytotoxic range of these drugs. Similar to the compounds 5c and 5h, the biscoumarin derivatives were also cytostatic, i.e., delayed cell proliferation and arrested cells in G_0_/G_1_ phase. This and our report add to the growing evidence of AChE inhibitors as potent antiproliferative agents [52,53,54,55]. Physostigmine, a reversible AChE inhibitor, for example, has been shown to reduce pancreatic cancer cell viability and invasion, macrophage infiltration, and pro-inflammatory signals through the inhibition of intrinsic AChE activity [56]. Abnormal expression of AChE has been observed in malignant grade III brain tumors [57], type IV glioma [58], lung cancer [59], retinoblastoma [60], squamous cell carcinoma [60], and human tumor cell lines of different tissue origin [61]. In many of these disorders, increased levels of AChE have been linked to increased metastatic potential of the tumor cells [59,62]. On the other hand, acetylcholine has also been suggested to act as a growth factor for lung epithelial cells, which is supported by the fact that its levels are elevated in lung cancer while that of AChE is decreased [63,64]. This would suggest that inhibition of AChE would promote cell proliferation and tumorigenesis, an effect that could be demonstrated through the use of AChE inhibitor eserine in a rat model of breast cancer [65,66]. This discrepancy might stem from the fact that different receptors were activated by acetylcholine; while activation of M3 muscarinic receptor has been shown to promote gastric and colon cancer progression, activation of M1 muscarinic receptor, on the other hand, inhibited pancreatic cancer cell proliferation [67,68].

Various other substitutions on the imidazo[1,2-*b*]pyridazine skeleton have also resulted in compounds that showed inhibition of cell proliferation, although the targets of action were different. These include the mitotic protein, haspin [69], interleukin-1 receptor-associated kinase 4 [70], inositol-requiring enzyme 1α kinase [71], dihydrofolate reductase [72], and MAPK-interacting kinase (MNK) [73]. However, the effect of these various substituent imidazo[1,2-*b*]pyridazine compounds on AChE activity has not been reported. Incidentally, several of these kinases are also inhibited by oxime compounds, which have been reported to have anticancer activities but are better known as AChE reactivators [74]. It must, however, be noted that the inhibition of kinases by oxime compounds is independent of its activity as an AChE reactivator, which refers to its ability to reverse the action of organophosphates on the enzyme [75].

Substituted imidazo[1,2-*b*]pyridazine compounds have been tested as drug candidates for acute toxoplasmosis [76], malaria [77,78], arthritis [79], epilepsy [80], and pain [81]. Their modes of action have been shown to involve inhibition of calcium-dependent protein kinase [76], p38 mitogen-activated protein kinase (MAPK) [79], nuclear factor κB (NF-κB) kinase subunit β (IKKβ)[82], mammalian target of rapamycin (mTOR) [83], protease-activated receptor 2 (PAR2) [84], vascular endothelial growth factor (VEGF) receptor 2 kinase [85], monopolar spindle 1 (Mps1) kinase [86], and Pim kinases [87], among others, depending on the structure of the synthesized inhibitor and the type of substitution on the imidazo[1,2-*b*]pyridazine skeleton. Although the added substituent differs in each case, the enzymes inhibited by the various substituted imidazo[1,2-*b*]pyridazine compounds are all associated with key pathways involved in cell proliferation and tumorigenesis. For example, inhibition of PAR2 by the imidazo[1,2-*b*]pyridazine compound, I-191, also leads to the downstream inhibition of p42/44 MAPK, calcium release, and cyclic adenosine monophosphate (cAMP) synthesis [84]. Physostigmine also mediates its antiproliferative activity on pancreatic cancer cells through the inhibition of p42/44 MAPK downstream of muscarinic receptors [56]. A substituted dihydropyridine ring on the imidazo[1,2-*b*]pyridazine skeleton shows inhibition of the Janus kinase member, tyrosine kinase 2, leading to downstream reduction of interferon synthesis [88]. Similarly, an (R)-2-phenylpyrrolidine substituted imidazopyridazine shows potent inhibition of the tropomyosin receptor kinase (Trk) family of proteins responsible for driving tumorigenesis in various cancer types [89]. The presence of a thiazolidinedione group together with the imidazopyridazine structure has been shown to be very effective against proliferation in various cancer cell types [90]. A piperazine substituent on the imidazo[1,2-*b*]pyridazine compound has been reported to inhibit phosphatidylinositol-4-kinase and cyclic guanidine monophosphate-dependent protein kinase of the malarial parasite Plasmodium [91]. Irrespective of the type of substitution present on the parent imidazo[1,2-*b*]pyridazine compound, these reports show that the signaling pathways leading to increased cell proliferation are the end-targets.

There is a need for the identification of newer generation AChE inhibitors that have additional properties including anti-cancer activity [92]. However seemingly different the phenomena of neurodegeneration and cancer are, many of the cancer signaling pathways also play a role in AD pathogenesis [93]. Although compounds 5c or 5h have not been tested for permeability across the blood–brain barrier, modifications that allow the compounds to cross have been reported for other imidazo[1,2-*b*]pyridazine compounds [94,95]. In this study, we also found that both compounds 5c and 5h increased mitochondrial stress (at 100 μM) and reduced intracellular ATP levels (significant at 10 μM). It is not known whether these compounds mediate oxidative stress through inhibition of any components on the mitochondrial electron transport system or by interfering with any of the intracellular pathways modulating reactive oxygen species levels in the cell. Nevertheless, the reduction in cellular ATP drives downstream neuroprotective measures such as phosphorylation of AMPK, a marker of mitochondrial dysfunction and a signal for the induction of autophagy [37,38]. We have previously demonstrated an increase in the levels of AMPK in IMR-32 cells in response to a complex I inhibitor, TaClo [20]. Increased mitochondrial dysfunction also leads to a shift in metabolism toward increased glycolytic dependence [20,96]. Studies by other groups on substituted imidazo[1,2-*b*]pyridazine compounds have not looked at their effect on cellular metabolism or mitochondrial stress. It is likely that the decrease in cellular ATP levels by compounds 5c and 5h drives cell cycle arrest and increased mitochondrial stress. The substituted imidazo[1,2-*b*]pyridazine compounds reported here, therefore, have multifunctional roles in AChE inhibition, anti-proliferation, and induced COX-2/iNOS inhibition.

## 4. Materials and Methods

### 4.1. Synthesis of the Compounds

The various steps involved in the synthesis of the compounds have been previously described [15]. Compound purity ≥99% was confirmed by both ^1^H-NMR and high-resolution mass spectrometer (HRMS) and was solubilized in dimethylsulfoxide (DMSO) at 100 mM concentration.

### 4.2. Cell Culture and Reagents

All reagents and kits were purchased from Sigma Aldrich (Bengaluru, India), HiChem Life Sciences (Ghaziabad, India), or Fisher Thermo Scientific India. 1-trichloromethyl-1,2,3,4-tetrahydro-β-carboline (TaClo) was provided from the lab of Prof. Gerhard Bringmann, University of Wurzburg, Germany. The mouse neuronal cell-line, Neuro-2A, the human neuroblastoma cell-line, IMR-32, and the human prostate cancer cell-line, DU-145, are not listed in the commonly misidentified cell line by the International Cell Line Authentication Committee (ICLAC; https://iclac.org/databases/cross-contaminations/; accessed on 14 December 2019 and also confirmed with the latest version released on 8 June 2021) and were obtained from the National Centre for Cell Science, Pune, India. These cells were cultured in Dulbecco’s MEM supplemented with 40 U/mL penicillin, 40 μg/mL streptomycin, 1 mM sodium pyruvate, 2 mM L-glutamine, and 10% heat-inactivated fetal calf serum (HiChem Life Sciences, Ghaziabad, India). Cells were cultured at a density of 5 × 10^4^ cells/cm^2^ in various formats (6-, 24-, or 96-well plates) as per experimental requirement and were stimulated the following day when the cells reached ~70% confluency. For most experiments, cells were stimulated with 10, 50, or 100 μM concentration of the compounds, and the controls were stimulated with the vehicle (DMSO) to a final concentration of 0.1%.

South Asian University Institutional Animal Ethics Committee (Reg. No. 1849/GO/Re/S/16/CPCSEA) approval was sought for use of animals for primary cell cultures as per the guidelines from the Committee for the Purpose of Control and Supervision of Experiments on Animals (CPCSEA). The study was carried out in compliance with the ARRIVE guidelines. Primary astroglial cell cultures were prepared from cerebral cortices of one-day-old neonatal Sprague-Dawley rats as previously described [41]. Cells were cultured in the same medium as above, and floating microglia were harvested every week (between 2–6 weeks) and re-seeded in 6- or 24-well plates to give pure microglial cultures. This method yields very pure microglia, which was confirmed by staining with CD11b. Culture preparations from two independent litters were used for the experiments.

### 4.3. Cell Viability and Apoptosis Assays

Cell viability was assayed in a 96-well format using 3-(4,5-dimethylthiazol-2-yl)-2,5-diphenyltetrazolium bromide (MTT) assay as described by the manufacturer (Sigma Aldrich, Bengaluru, India). Briefly, 5000 cells were plated per well of a 96-well plate and incubated with different concentrations of the compounds for 24 h. Absorbance was measured at 550 nm in a microplate reader (Synergy HT Biotek, Biotek Instruments, New York, NY, USA). The absorbance value obtained for untreated cells was considered to be 100%. Eight wells per condition were used, and the experiments were repeated thrice using cells from different passages.

Cell viability was also measured through DNA fragmentation analysis as described elsewhere [97]. Ethanol-fixed cells were permeabilized with 0.1% triton X-100 followed by staining with propidium iodide (PI, 20 μg/mL). PI-positive cells were then counted on a flow cytometer that correlated to apoptotic cells.

The percentage of cells undergoing apoptosis was also measured using the PI-Annexin V staining method as per the manufacturer’s protocol (Thermo Fisher Scientific, Mumbai, India). Briefly, cells stimulated with different compounds were washed twice in ice-cold phosphate-buffered saline (PBS) and resuspended in a buffer comprising 10 mM HEPES/NaOH, pH 7.4, 140 mM sodium chloride (NaCl), and 2.5 mM calcium chloride (CaCl_2_) at a concentration of 1 × 10^6^ cells/mL. A total of 100 μL cell suspension was aliquoted and stained with PI and Annexin V for 15 min at RT. A total of 30,000 cells were counted on BD FACS VerseTM flow cytometer and analyzed using BD FACS Suite software (BD Biosciences, San Jose, CA, USA).

### 4.4. Cell Cycle Analysis

Cell cycle was measured using a PI-based method in ethanol-fixed cells that have been stimulated with the compounds for 24 h as described previously [20]. Briefly, cells were fixed in ice-cold 80% ethanol followed by two washes in PBS and digestion with RNaseA (50 U/mL) for one hour. Cells were then stained with PI (20 μg/mL) and analyzed on a BD FACS VerseTM flow cytometer. A minimum of 30,000 events were collected for each sample and analyzed using ModFit LT software (Verity Software House, Topsham, ME, USA).

### 4.5. Cell Proliferation Assays

The anti-proliferative effect of imidazo[1,2-*b*]pyridazine compounds was assessed through a colony formation assay. For this, IMR-32 cells were cultured in a 6-well format at a low density of 500 cells/well. After the formation of initial colonies in 2–3 days, the cells were stimulated with different compounds [32]. Five days post-stimulation, cells were washed twice with PBS and then fixed with 4% paraformaldehyde for 10 min. After two washes with PBS, cells were stained with 0.2% (w/v) crystal violet for 10 min. Excess dye was removed by multiple washings in PBS and dried at room temperature. The plates were imaged at 10X and parameters such as cell count, total area covered, and integrated density of the colonies were quantified using NIH Image J software (https://imagej.nih.gov/ij, accessed on 20 March 2019).

Cell proliferation was also measured using the CyQUANT method (Thermo Fisher Scientific, Mumbai, India), which employs a DNA-binding fluorescent dye to quantify the total number of cells in the well [33]. Briefly, 5000 cells per well were seeded in a 96-well format and stimulated with different compounds for 24 h followed by washing twice with PBS and addition of an equal volume of dye-binding solution containing CYQUANT-NF reagent. Cells were incubated for 40 min at 37 °C, and fluorescence intensity was measured at an excitation wavelength of 485 nm and emission of 530 nm using a fluorescence microplate reader (Synergy HT Biotek, Biotek Instruments, New York, NY, USA). A standard curve was also made using known cell numbers (0–20,000) for comparative analysis.

### 4.6. Cell Migration Assay

Migration of IMR-32 cells was studied using the Boyden chamber assay (Corning, Gurugram, India) [39]. IMR-32 cells were seeded (5000 cells/insert) on the upper chamber of a Transwell insert in serum-free media, and serum-containing media were added in the lower chamber of the Transwell. The media in the lower chamber contained the respective treatment (10 and 100 μM each of compounds 5c and 5h). Cells were incubated in a humidified 5% CO_2_ incubator kept at 37 °C. After 24 h, cells on the upper chamber were scrapped while the migrated cells from the lower chamber were fixed in 70% ethanol and stained with 1 mg/mL Hoechst 44432 for 5 min. The stained cells were imaged under a fluorescence microscope and counted using the NIH Image J software (https://imagej.nih.gov/ij, accessed on 11 January 2021).

### 4.7. Western Blotting

Total cell lysates were prepared in a sodium dodecylsulfate (SDS) lysis buffer composed of 42 mM Tris-HCl, pH 6.8, 1.3% (w/v) SDS, 6.5% glycerol, 0.1 mM sodium orthovanadate, and protease inhibitor cocktail (Sigma Aldrich, Bengaluru, India), as described previously [20,41]. Protein content was measured using the bicinchoninic acid method (Thermo Fisher Scientific, Mumbai, India), using bovine serum albumin (BSA) as standard. 2-mercaptoethanol (1% final concentration) and bromophenol blue (0.2 mg/mL) were added to the samples and heated at 95 °C for 5 min before electrophoresis. A total of 30 μg samples were loaded on a 7.5% (for AMPK, iNOS, and COX-2), 12% (for p27^Kip1^), or 15% (for cleaved caspase 3) polyacrylamide gel under reducing conditions. Separated proteins were transferred onto a polyvinylidene fluoride membrane (Merck LifeSciences, Mumbai, India) and blocked for one hour with 5% BSA in Tris-buffered saline containing 0.1% Tween-20 (TBS-T). Primary antibodies used were rabbit anti-phospho-AMPKα (detecting endogenous levels of AMPKα only when phosphorylated at Thr172), rabbit anti-AMPKα, and rabbit anti-cleaved caspase 3 (detecting endogenous levels of the large fragment p17/p19 of the activated caspase 3, all from Cell Signaling Tech, Danvers, MA, USA), rabbit anti-inducible nitric oxide synthase (Cayman Chemicals, Ann Arbor, MI, USA), rabbit anti-p27^Kip1^ (Santa Cruz Biotechnology, Dallas, TX, USA), and mouse anti-COX-2 (Santa Cruz Biotechnology or Cayman Chemicals), all diluted at 1:1000 in TBS-T containing 1% BSA overnight at 4 °C. For the normalization of protein loaded, mouse anti-β-actin (Sigma Aldrich, Bengaluru, India) was used at 1:5000 dilution. Secondary antibody was diluted 1:10000 (for both mouse and rabbit) in 1% BSA in TBS-T for 1 h at room temperature and washed extensively. Proteins were detected using a chemiluminescent solution made by mixing equal volumes of solution A (2.5 mM luminol, 0.396 mM p-coumaric acid, and 0.1M Tris-HCl, pH 8.5) and solution B (5.2mM H_2_O_2_ and 0.1 M Tris-HCl pH 8.5).

### 4.8. Mitochondrial Oxidative Stress Analysis

Mitochondrial oxidative stress was measured by staining with a mitochondria-specific dye, Mito-SOX Red (Thermo Fisher Scientific, Mumbai, India), using both confocal microscopy and flow cytometry as previously described [96]. For flow cytometric analysis, cells treated with different compounds were washed twice with PBS and then resuspended in 0.1 mL PBS containing Mito-SOX Red at a concentration of 5 μM for 15 min at RT. Cells were then diluted with 0.9 mL PBS and 50,000 cells were counted on the flow cytometer and analyzed using BD FACS Suite software (BD Biosciences, San Jose, CA, USA). For confocal microscopy, cells were cultured on coverslips and treated with different compounds for 24 h. Media were then replaced with serum-free media, and cells were stained with 5 μM Mito-SOX for 15 min. Cells were then washed in PBS, fixed with 4% paraformaldehyde, and stained with the nuclear dye, Hoechst 33342. The coverslips were mounted on a glass slide and imaged at 100X magnification using a confocal microscope (Nikon Eclipse, Nikon Technologies, Tokyo, Japan) with an excitation wavelength of 510 nm and emission of 580 nm. Image analysis was performed using NIH Image J software (https://imagej.nih.gov/ij, accessed on 5 March 2020).

Cellular reactive oxygen species were also detected using the cell-permeant reagent, H_2_DCFDA (Thermo Fisher Scientific, Mumbai, India). IMR-32 cells were treated with the compounds for 24 h, and then the media was replaced with PBS to minimize extracellular hydrolysis of the reagent. A total of 10 μM H_2_DCFDA reagent was added to the cells and incubated for 20 min. Cells were washed and then read in a microplate reader (Biotek Instruments, New York, NY, USA) with an excitation wavelength of 485 nm and emission wavelength of 528 nm.

### 4.9. Cellular ATP and Released Nitrate Estimation

Total cellular ATP levels were measured using a bioluminescent ATP assay kit (Promega, New Delhi, India) as per the manufacturer’s instructions. Luminescence was recorded with a microplate reader (Biotek Instruments, New York, NY, USA). For the quantitation of total nitric oxide synthesized by cells stimulated with different compounds for 24 h in the presence or absence of LPS, an assay measuring total nitrate released in the culture media was used. This colorimetric assay was performed using a commercially available kit as per the manufacturer’s instructions (Cayman Chemicals, Ann Arbor, MI, USA).

### 4.10. Statistical Analysis

All experiments were performed in triplicates and used cultures from at least three different passages. For the primary cell cultures, data were obtained from two separate preparations using cells obtained from litters of different parents. All data were analyzed using Microsoft Excel 2011. Data were presented as mean ± SEM and a Shapiro-Wilk test was used for determining the normality of the data. Statistical significance was calculated using a Student’s *t*-test. The effect of each compound was compared with that of the control (vehicle)-treated cells (paired comparison). Analysis of variance (ANOVA) was used when calculating the significance of different cell cycle phases in cells treated with the compounds. Differences were considered significant when *p* < 0.05.

## 5. Conclusions

In summary, the 3-nitro-6-amino-imidazo[1,2-*b*]pyridazine compounds shown in this paper have piperidine substituent in compound 5c and 1-phenylpiperazine substituent in compound 5h. Both compound 5c and 5h inhibited cell proliferation, with compound 5h exhibiting a more potent effect including an increase in the activity of caspase 3. Both compounds arrested cells in G_0_/G_1_ phase, reduced intracellular ATP levels, and enhanced mitochondrial oxidative stress. In addition, both compounds significantly inhibited LPS-induced COX-2 and iNOS expression in primary microglial cells. Inhibition of COX-2 and AChE is a desirable feature of an anti-cancer drug considering the role of these enzymes in cancer cell progression and metastasis. We report for the first time that the substitutions made on the parent imidazo[1,2-*b*]pyridazine compound resulted in a multifunctional lead molecule that shows AChE inhibition and also exhibits anti-proliferative and anti-inflammatory properties. The suitability of these compounds for therapeutic potential should be exploited further.

## Figures and Tables

**Figure 1 molecules-26-05319-f001:**
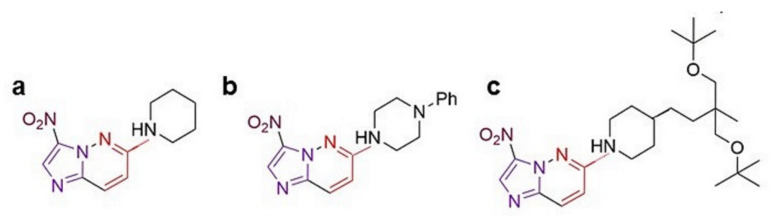
The structure of the compounds used in this study: (**a**) compound 5c, (**b**) compound 5h, and (**c**) compound 5k. Synthesis of these compounds has been described previously [15].

**Figure 2 molecules-26-05319-f002:**
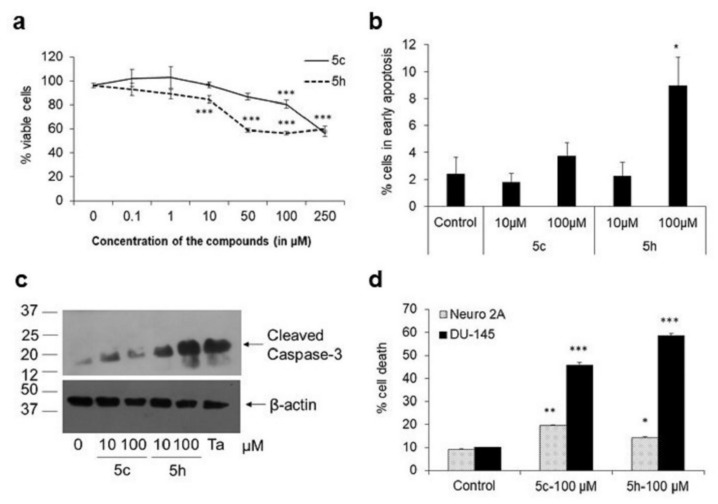
Substituted imidazo[1,2-*b*]pyridazine compounds show differential cytotoxicity. (**a**) IMR-32 cells were stimulated with compounds 5c and 5h at different doses for 24 h followed by MTT assay. Relative viability of the cells in comparison to control (set to 100%) is depicted. Only vehicle (DMSO) is used at concentration 0. Eight wells per condition were used and at least four independent experiments were conducted. *** *p* < 0.001, with respect to untreated cells. Error bars represent SEM. (**b**) Cells treated with compounds 5c or 5h for 24 h were stained with PI and Annexin V and analyzed on a flow cytometer. The percentage of total cells that were PI negative and Annexin V positive are depicted here as cells in early apoptosis. * *p* < 0.05, with respect to untreated cells. Error bars represent SEM, *n* = 4. (**c**) IMR-32 cells treated with 10 or 100 μM compound 5c or 5h were harvested after 24 h for analysis of cleaved caspase 3 levels through Western blot. As a positive control, cells were also stimulated with 250 μM TaClo, which is known to activate caspase 3 in neuroblastoma cells. (**d**) The mouse neuronal cell-line, Neuro-2A, and the human prostate cancer cell line, DU-145, were treated with 100 μM compound 5c or 5h for 36 h. Cells were then stained with PI and viable cells analyzed on a flow cytometer. * *p* < 0.05, ** *p* < 0.01, *** *p* < 0.001, with respect to control cells. Error bars represent SEM.

**Figure 3 molecules-26-05319-f003:**
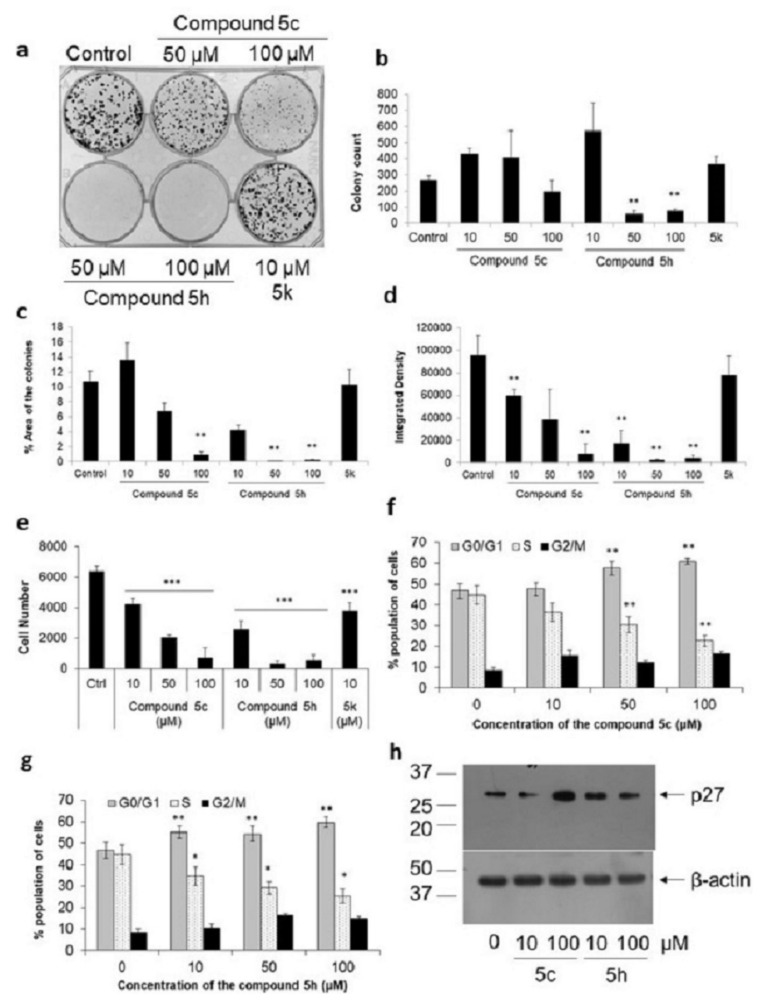
Substituted imidazo[1,2-*b*]pyridazine compounds affect IMR-32 cell proliferation. Five hundred cells/well were plated in a 6-well plate and after 48 h stimulated with various compounds. Five days post-stimulation, the colonies were fixed and stained with 0.2% crystal violet. (**a**) A representative image of a plate is depicted here. Using Image J, the total number of colonies (**b**), area covered by the colony (**c**), and integrated density of the colonies (**d**) were calculated as described in Methods. (**e**) IMR-32 cells were plated in a 96-well plate (eight wells per condition) and stimulated with different compounds. After 24 h, CyQUANT NF dye was added and fluorescence measured. (**f**,**g**) IMR-32 cells were stimulated with different concentrations of the compound 5c (**f**) or 5h (**g**) for 24 h. Cells were fixed and stained with PI, and at least 30,000 cells were counted and analyzed on a flow cytometer. Percentage of total cells in each of the G_0_/G_1_, S, and G_2_/M phases are depicted here. (**h**) IMR-32 cells were stimulated with the compounds for 24 h. Cell lysate was collected for the analysis of p27^Kip1^ protein through Western blot. β-actin was used for normalization as described in Materials and Methods. * *p* < 0.05, ** *p* < 0.01, and *** *p* < 0.001, with respect to control cells. Error bars represent SEM, *n* = 3 independent experiments.

**Figure 4 molecules-26-05319-f004:**
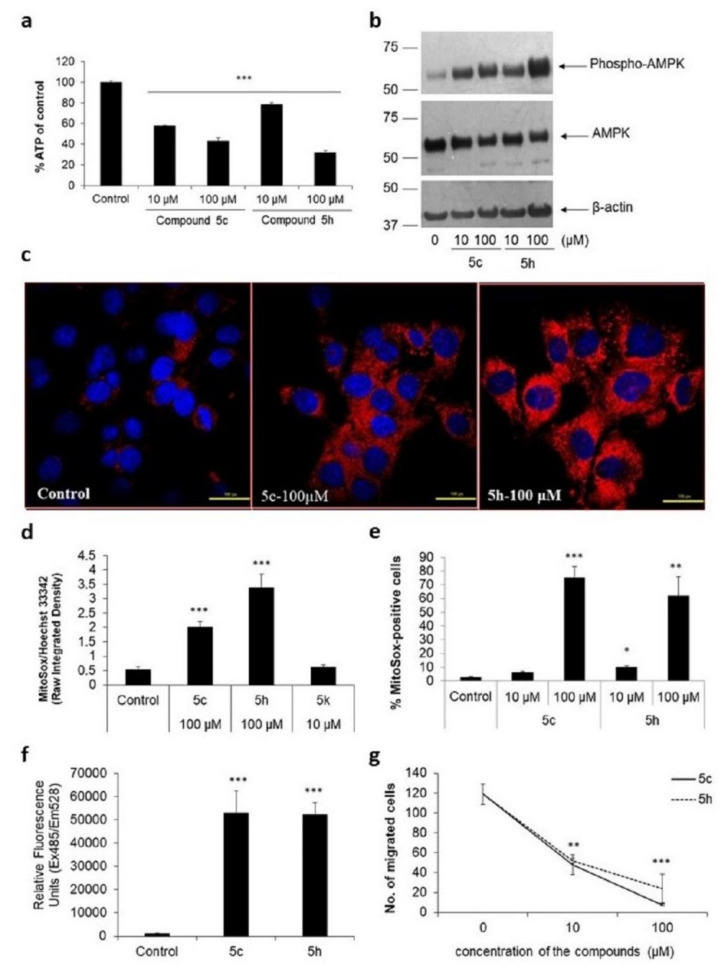
Compounds 5c and 5h mediate oxidative stress. (**a**) IMR-32 cells were stimulated with different concentrations of compounds 5c and 5h for 24 h in a 96-well format. Media were then removed and total ATP levels in each well were then measured using a bioluminescent kit. The relative luminescent units of untreated cells were considered to have a value of 100% to which stimulated cells were compared. *** *p* < 0.001 with respect to control cells. Error bars represent SEM, *n* = 3. (**b**) IMR-32 cells were stimulated with different doses of compounds 5c and 5h in a 6-well format, and total cell lysates were prepared after 24 h as described in Methods. The lysates were analyzed through Western blotting to show increased levels of phosphorylated AMPK protein. (**c**) IMR-32 cells were plated on coverslips and stimulated with 100 μM compounds 5c or 5h for 24 h. Cells were then washed, incubated with Mito-Sox Red, and fixed as described in Methods. A representative image under a confocal microscope is depicted here (scale bar, 10 μm). Mito-Sox Red is depicted in red, while Hoechst 33342, a counterstain used to detect nuclear DNA, is depicted in blue. (**d**) A minimum of 10 fields per coverslip were imaged and analyzed on Image J software. Raw integrated density, the sum of all the pixels in the image, was calculated for Mito-Sox Red signal and normalized with that from the Hoechst 33342 signal. *** *p* < 0.001 with respect to control cells. (**e**) Mito-Sox Red-positive IMR-32 cells, which were stimulated with compounds 5c or 5h for 24 h, were also analyzed on a flow cytometer with at least 30,000 events counted per condition. * *p* < 0.1, ** *p* < 0.01, and *** *p* < 0.001, with respect to control cells. Error bars represent SEM, *n* = 3 independent experiments. (**f**) Reactive oxygen species were also detected in IMR-32 cells stimulated with 100 μM compound 5c or 5h for 24 h using the cell-permeant dye, H_2_DCF-DA. Increase in fluorescence was read in a multimode microplate reader set with excitation 485 nm and emission 528 nm. *** *p* < 0.001 with respect to control cells. (**g**) Cell migration was tested using the Boyden chamber assay. IMR-32 cells were plated on top of the Transwell insert and stimulated with the compounds. The following day cells that have migrated to the bottom of the insert were stained with Hoechst 44432 and counted on a microscope. ** *p* < 0.01 and *** *p* < 0.001, with respect to control cells. Error bars represent SEM, *n* = 3 independent experiments.

**Figure 5 molecules-26-05319-f005:**
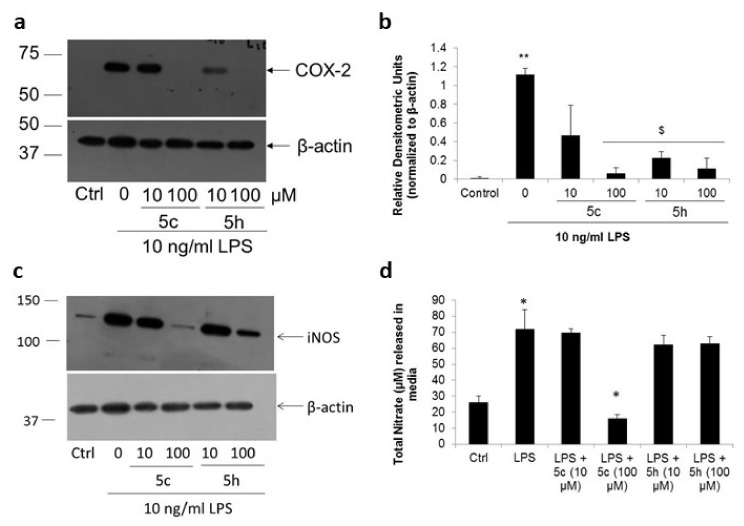
Substituted imidazo[1,2-*b*]pyridazine compounds inhibit pro-inflammatory enzymes. (**a**,**b**) Primary rat microglial cells were stimulated with 10 ng/mL LPS in the presence or absence of compounds 5c or 5h for 24 h. Representative Western blot of COX-2 (**a**) and relative densitometric analysis shows both compounds 5c or 5h significantly inhibited LPS-induced COX-2 in microglial cells (**b**). ** *p* < 0.01 with respect to control cells, ^$^
*p* < 0.05 with respect to LPS-treated cells. Error bars represent SEM, *n* = 3. (**c**) Representative Western blot for inducible nitric oxide synthase (iNOS). (**d**) Media supernatants were collected 24 h post-stimulation for the measurement of nitric oxide released in the culture media. * *p* < 0.05 with respect to control or LPS-treated cells.

## Data Availability

All the datasets that have been used in the current study are available from the corresponding author upon request.

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
