# Peer review of "Imidazopyridazine Acetylcholinesterase Inhibitors Display Potent Anti-Proliferative Effects in the Human Neuroblastoma Cell-Line, IMR-32"

_molecules, 2021, doi:10.3390/molecules26175319_

Round 1

Reviewer 1 Report

The manuscript of Sharma et al is focused on the investigation of toxicity of two compounds derivatives of Imidazo[1,2 -b]pyridazine. The authors showed that compounds 5c and 5h, that previously showed inhibitory activity towards Acetylcholinesterase (AChE), has toxic, antiproliferative and anti-inflammatory effects in neuroblastoma cell line IMR-32. As a drug development for neurodegenerative disorders treatment stays an important problem the manuscript is of scientific interest. The study is logically structured, and the results are presented in well organized manner.  However, I have some concerns regarding the manuscript.

  1. The introduction and the discussion parts are not that much well organized. Hard to understand the main goal/goals of the study. The introduction part as well as discussion part seem to be a bit all over the place: the reader cannot catch clearly what was the goal of the authors. First, one understands that compounds tested have the potential as a drug for AD treatment due to the inhibition of AChE and the neuroblastoma cell line is used as it is a model object for the study of AD to test the toxic effects. After the authors switch to the potential of the drug in the tumors treatment. This is very confusing. It seems that it should be better to stick to the one common line in the paper and state clearly what was the goal.
  2. No results shown in the manuscript for the concentration of the compounds close to the IC50. There is no even note about this concentration before the discussion part, where it is stated that low concentrations were not toxic. It should be understood from the experiments with higher concentration, but it would be nice to add the description of the choice of concentrations tested.
  3. In the experiment to test cytotoxic effect the authors have chosen the different time point of 36 h for the murine neuronal cell line neuro A2 and prostate cancer cell line DU 145. It is not clear how it corresponds to the toxicity after 24 h for IMR-32 cells. Were the compounds toxic after 24 hours in neuro A2 and DU 145 as well?
  4. The authors used for the mito SOX red to estimate mitochondria induced oxidative stress. I would suggest using at least one more method to confirm oxidative stress induction (dihydroethidium e.g.) as ROS estimation known to be tr.icky However it is up to authors.
  5. It is not clear why the authors used Student’s t test for multiple group comparison. Appropriate statistical test should be used.
  6. The conclusion about “enhanced AChE inhibition and having anti-cancer properties” can not be based on the results presented in the manuscript. It is too strong and should be removed and rewritten in more suggestive way.

Overall impression of the manuscript was dumped a bit by the lack of the strict line of presentation of the conclusions regarding the discovered properties of the compounds and their potential application.

Author Response

We thank the Reviewer for appreciating our study and for providing encouraging and useful comments. We have tried to incorporate all the suggestions mentioned by the Reviewer. We believe this has strengthened the manuscript further and look forward to the views of the Reviewer.

  1. The introduction and the discussion parts are not that much well organized. Hard to understand the main goal/goals of the study. The introduction part as well as discussion part seem to be a bit all over the place: the reader cannot catch clearly what was the goal of the authors. First, one understands that compounds tested have the potential as a drug for AD treatment due to the inhibition of AChE and the neuroblastoma cell line is used as it is a model object for the study of AD to test the toxic effects. After the authors switch to the potential of the drug in the tumors treatment. This is very confusing. It seems that it should be better to stick to the one common line in the paper and state clearly what was the goal.

Response to the Reviewer: We have now incorporated a sentence describing the goal of the study in the Introduction section. We have also reorganized the Discussion to elucidate the purpose of the study. The Discussion begins with the AChE inhibitory activity of these compounds. The results of this study on cytotoxicity and cell proliferation brings the observation of the utility of AChE inhibitors as antiproliferative drugs. The discussion then follows the role of acetylcholine in systemic body, results of other AChE inhibitors in cell proliferation, molecular targets of other substitutions on imidazopyridazine compounds and their role in cell proliferation. We hope the reorganized Discussion is clearer. The Graphical Abstract depicts the hypothetical pathway within cells stimulated with the substituted imidazo[1,2-b]pyridazine compounds.

  1. No results shown in the manuscript for the concentration of the compounds close to the IC50. There is no even note about this concentration before the discussion part, where it is stated that low concentrations were not toxic. It should be understood from the experiments with higher concentration, but it would be nice to add the description of the choice of concentrations tested.

Response to the Reviewer: We had tested the compounds at 0.1 μM and 1 μM doses where the reduction of MTT was similar to control cells. As no effect was seen below 10 μM, we had not included this in the earlier version. However, this data has now been included in the revised Results section and Figure 2a has been accordingly depicted.

  1. In the experiment to test cytotoxic effect the authors have chosen the different time point of 36 h for the murine neuronal cell line neuro A2 and prostate cancer cell line DU 145. It is not clear how it corresponds to the toxicity after 24 h for IMR-32 cells. Were the compounds toxic after 24 hours in neuro A2 and DU 145 as well?

Response to the Reviewer: A personal emergency of the student responsible for the experiment led to a longer incubation of the cells up to 36 h for cell death assay using propidium iodide. Accordingly, the student performed the replicate experiments also at 36 h. Although we planned to later redo this experiment at 24 h, an unexpected lockdown and the resulting disruption in resources and manpower prevented us from repeating at 24 h. The data that we have of 36 h is therefore presented here. We reconfirmed that the compounds were toxic at 24 h, similar to IMR-32 cells, through MTT assay.

  1. The authors used for the mito SOX red to estimate mitochondria induced oxidative stress. I would suggest using at least one more method to confirm oxidative stress induction (dihydroethidium e.g.) as ROS estimation known to be tricky However it is up to authors.

Response to the Reviewer: We had used two different methods – confocal microscopy and flow cytometry – to confirm the results of Mito-SOX Red. Mito-SOX Red dye uses dihydroethidium as a reactive oxygen species indicator. Dihydroethidium can detect reactive oxygen species within the cytosol and its oxidized form intercalates with the cellular DNA staining the nucleus bright red. Therefore, for specific detection of oxidative stress within the mitochondria, dihydroethidium is conjugated to a positively charged, lipophilic triphenylphosphonium cation (TPP+) which is mitochondria-specific. This conjugate of TPP+-dihydroethidium is sold as Mito-SOX Red.

Nevertheless, we used another method to confirm oxidative stress in cells treated with the compounds as suggested by the Reviewer. We used the cell permeant reagent, 2’,7’-dichlorodihydrofluorescein diacetate (H2DCFDA) which is oxidized by reactive oxygen species to the highly fluorescent, 2’,7’-dichlorofluorescein which can be easily measured through a multimode plate reader (new Figure 4f in the revised manuscript). We hope the Reviewer is satisfied with the data on oxidative stress obtained through two different indicator dyes.  

  1. It is not clear why the authors used Student’s t test for multiple group comparison. Appropriate statistical test should be used.

Response to the Reviewer: All data shown in this manuscript are paired comparisons only between control (vehicle) and compound (either compound 5c or 5h)-treated cells. Since the effect of each compound was compared with untreated cells, we performed Student’s t-test. We have clarified this in the Methods section of the revised manuscript. We have used ANOVA in the newly revised Figure 3f and 3g, where we were comparing the effect of the compounds in different cell cycle phases.

6. The conclusion about “enhanced AChE inhibition and having anti-cancer properties” can not be based on the results presented in the manuscript. It is too strong and should be removed and rewritten in more suggestive way.

Response to the Reviewer: We have rewritten the statement as suggested by the Reviewer in the revised manuscript.

Overall impression of the manuscript was dumped a bit by the lack of the strict line of presentation of the conclusions regarding the discovered properties of the compounds and their potential application.

Response to the Reviewer: We hope the revised manuscript explains the new properties of these compounds and its application in a better manner. The Graphical Abstract also depicts the schematic of the targets identified in this project.

Reviewer 2 Report

The use of AChE inhibitors as anti-proliferative agents is interesting, but I think that in the introduction the authors have to better explain the role of this enzyme in neuroblastoma cells. The english language should be improve, there are many errors, for example in line 105 replace "substitution" with "substituent", in line 298 replace "piperadine" with "piperidine" and many others. The title of the article is not appropriate because the molecules are not new so the adjective "novel" can not be used. Finally, the authors have to delete the chemistry paragraph, figure 1 and supplementary files, because these molecules have been synthesized previously. I suggest to insert only the structures of the compounds studied in this work (5c and 5h) with the appropriate reference.   

Author Response

1. The use of AChE inhibitors as anti-proliferative agents is interesting, but I think that in the introduction the authors have to better explain the role of this enzyme in neuroblastoma cells.

Response to the Reviewer: We thank the Reviewer for the positive comments provided. As per the suggestion given, we have added a few additional lines on the role of AChE in neuroblastoma cells in the Introduction.

2. The english language should be improve, there are many errors, for example in line 105 replace "substitution" with "substituent", in line 298 replace "piperadine" with "piperidine" and many others.

Response to the Reviewer: We sincerely apologize for the many errors and hope the revised version has greatly improved the manuscript.

3. The title of the article is not appropriate because the molecules are not new so the adjective "novel" can not be used.

Response to the Reviewer: The substituted imidazo[1,2-b]pyridazine compounds described in this study are relatively new. This is the first study demonstrating the biological efficacy of these compounds. However, as per the Reviewer’s suggestion, we have replaced the word “novel” with “imidazopyridazine” to denote the key structural molecule studied in this report. The revised title is “Imidazopyridazine acetylcholinesterase inhibitors display potent anti-proliferative effects in the human neuroblastoma cell-line, IMR-32” which we believe suits the work.

4. Finally, the authors have to delete the chemistry paragraph, figure 1 and supplementary files, because these molecules have been synthesized previously. I suggest to insert only the structures of the compounds studied in this work (5c and 5h) with the appropriate reference. 

Response to the Reviewer: As per the suggestion of the Reviewer, we have now removed the chemistry part including the supplementary file and have remade Figure 1 to depict only the structures of the compounds used in this study. We sincerely thank the Reviewer for all the suggestions provided which we hope has strengthened our manuscript.

Reviewer 3 Report

  1. Cell cycle was measured using PI-based method. This method allows you to distinguish cell cycle phases according to DNA content – G0/G1, S and G2/M – phases, but not G0 and G1 – phases. Please correct G1 to G0/G1 – lines 20, 203, 310, 592 and Figure 3f.
  2. Only G0/G1-phase data are presented. What about S and G2/M – phases? These drugs are suggested as anticancer cytostatics for treatment. In this case, a complete analysis of the cell cycle carries important information for understanding cell-cycle checkpoints mechanisms. Please provide the complete cell-cycle analysis data (including S and G2/M data).

Author Response

  1. Cell cycle was measured using PI-based method. This method allows you to distinguish cell cycle phases according to DNA content – G0/G1, S and G2/M – phases, but not G0 and G1 – phases. Please correct G1 to G0/G1 – lines 20, 203, 310, 592 and Figure 3f.

Response to the Reviewer: We thank the Reviewer for pointing out this mistake. While we have been very careful to always use the term “G0/G1” in the manuscript, at certain places we missed out. We sincerely apologize for this oversight and have now corrected it across the manuscript in the revised version.

  1. Only G0/G1-phase data are presented. What about S and G2/M – phases? These drugs are suggested as anticancer cytostatics for treatment. In this case, a complete analysis of the cell cycle carries important information for understanding cell-cycle checkpoints mechanisms. Please provide the complete cell-cycle analysis data (including S and G2/M data).

Response to the Reviewer: We have provided complete cell cycle analysis in the revised manuscript in the figures 3f and 3g. There is a significant increase in the cells in G0/G1 phase as indicated in the manuscript and also a significant decrease in cells in S phase which we have written in the revised version. However, no significant change in population of cells in G2/M phase was observed. This indicated that there is a dysregulation at the G1/S transition. In addition, we have also included a new data on the negative cell cycle regulator protein, p27Kip1, whose levels rise during the G0/G1 phase and is clearly visible in cells treated with the compounds (Figure 3h in the revised manuscript). We hope this provides additional confirmation of cell cycle arrest in IMR-32 cells treated with substituted imidazo[1,2-b]pyridazine compounds.

Round 2

Reviewer 2 Report

I think that this manuscript is now suitable for the pubblication in this journal.